# Strategies to Enhance Diagnostic Capabilities for the New Drug-Resistant Tuberculosis (DR-TB) Drugs

**DOI:** 10.3390/pathogens13121045

**Published:** 2024-11-28

**Authors:** Antonia Morita Iswari Saktiawati, Anca Vasiliu, Francesca Saluzzo, Onno W. Akkerman

**Affiliations:** 1Department of Internal Medicine, Faculty of Medicine, Public Health, and Nursing, Universitas Gadjah Mada, Yogyakarta 55584, Indonesia; 2Center for Tropical Medicine, Faculty of Medicine, Public Health, and Nursing, Universitas Gadjah Mada, Yogyakarta 55584, Indonesia; 3Global TB Program, Department of Pediatrics, Baylor College of Medicine, Houston, TX 77030, USA; anca.vasiliu@bcm.edu; 4Clinical Infectious Diseases, Research Center Borstel, Leibniz Lung Center, 23845 Borstel, Germany; 5Clinical Tuberculosis Unit, German Center for Infection Research (DZIF), Ham-burg-Lübeck-Borstel-Riems, 23845 Borstel, Germany; 6IRCCS San Raffaele Scientific Institute, 20132 Milan, Italy; saluzzo.francesca@hsr.it; 7Department of Pulmonary Diseases and Tuberculosis, University Medical Center Groningen, University of Groningen, 9713 GZ Groningen, The Netherlands; o.w.akkerman@umcg.nl; 8Tuberculosis Center Beatrixoord, University Medical Center Groningen, University of Groningen, 9751 ND Haren, The Netherlands

**Keywords:** diagnosis, drug resistance, tuberculosis, microbial sensitivity tests, antitubercular agents, global health

## Abstract

The global burden of drug-resistant tuberculosis (DR-TB) continues to challenge healthcare systems worldwide. There is a critical need to tackle DR-TB by enhancing diagnostics and drug susceptibility testing (DST) capabilities, particularly for emerging DR-TB drugs. This endeavor is crucial to optimize the efficacy of new therapeutic regimens and prevent the resistance and overuse of these invaluable weapons. Despite this urgency, there remains a lack of comprehensive review of public health measures aimed at improving the diagnostics and DST capabilities. In this review, we outline strategies to enhance the capabilities, especially tailored to address the challenges posed by resistance to new DR-TB drugs. We discuss the current landscape of DR-TB drugs, existing diagnostic and susceptibility testing methods, and notable gaps and challenges in these methods and explore strategies for ensuring fair access to DST while narrowing these disparities. The strategies include public health interventions aimed at strengthening laboratory infrastructure, workforce training, and quality assurance programs, technology transfer initiatives, involving drug developers in the DST development, establishing national or regional referral hubs, fostering collaboration and resources pooling with other infection control efforts, extending testing access in underserved areas through public–private partnerships, advocating for lowering costs or loans at low interest, remote technical support, and implementing mandatory molecular surveillance monitoring. This review underscores the urgent need to enhance DST capacities for new DR-TB drugs and identifies opportunities for innovation and improvement. Assessing the extent of the global health impact of these measures is crucial to ensure their effectiveness in combating DR-TB.

## 1. Introduction

Multidrug-resistant tuberculosis (MDR-TB) is a significant public health concern characterized by resistance to at least two of the most powerful anti-TB drugs: isoniazid and rifampicin. This form of TB presents a serious challenge for global TB elimination efforts, as it necessitates prolonged and more complex treatment regimens that are less effective and more expensive than those for drug-susceptible TB.

In 2022, the World Health Organization (WHO) estimated that approximately 410,000 individuals (95% uncertainty interval (UI): 370,000–450,000) contracted multidrug-resistant or rifampicin-resistant tuberculosis (MDR/RR-TB) [1]. The prevalence of MDR/RR-TB varies significantly across regions and countries. The regions with the highest burden are Southeast Asia (45%), Europe (22%), Western Pacific (18%), Africa (12%), the Americas (3%), and the Eastern Mediterranean (3%) [1]. Analysis of survey data from 156 diverse settings reveals a prevalence of isoniazid-resistant TB at 7.4% and 11.4% among individuals who have never been treated for TB and those who have received previous treatment, respectively [2]. Among people with bacteriologically confirmed pulmonary TB, 73% were tested for rifampicin resistance, and among those tested, 4.4% were diagnosed with rifampicin-resistant TB (RR-TB) [1]. Only 43% of the estimated number of people who develop MDR/RR-TB receive treatment, and the rate of treatment success for MDR/RR-TB has gradually improved, reaching approximately 63%, up from approximately 50% in treated cases in 2012 [1].

New drug regimens incorporating bedaquiline, clofazimine, linezolid, pretomanid, and delamanid to treat MDR and (pre)-extensively (XDR)-TB have been recommended by the WHO and are being implemented globally [3]. These regimens have a shorter duration for treating MDR/RR-TB and pre-XDR-TB, a better safety profile [4], and better treatment outcomes than the old regimens [5,6,7]. Nevertheless, the use of these regimens could lead to the rapid development of drug resistance, making effective treatments less available and wasting these new and repurposed drugs [8,9,10,11]. Therefore, detecting resistance to new anti-TB drugs is of critical importance in the effective management and control of MDR/RR-TB.

Culture-based phenotypic (p)DST is still considered the gold standard for determining resistance to these new drug regimens [12,13], but it is technically challenging, costly, and time-consuming. Meanwhile, the new type of nucleic acid amplification tests (NAATs) can predict resistance only to fluoroquinolones, second-line injectable drugs, and ethionamide [14]. Today, only next-generation sequencing (NGS) can predict resistance to all new TB drugs, but it needs high technical and analytical skills, storage size, and security [15]. Worldwide, there is unequal access to DST due to several factors, such as limited infrastructure in low- and middle-income countries (LMICs) [15] and higher cost of sequencing facilities in LMICs than in high-income countries [16].

Despite the urgency to enhance diagnostic and DST capabilities, there remains a lack of comprehensive reviews on public health measures aimed at improving these capacities, particularly for new drugs. Therefore, this review aims to provide insights into the multifaceted approaches necessary to increase diagnostic capacities for the new DR-TB drugs and to reduce the gap in DST access worldwide, thereby contributing to the global efforts to combat this formidable public health threat.

## 2. Current Drugs for DR-TB

The official WHO guidelines on MDR/RR-TB until 2019 advised treatment for MDR/RR-TB to consist of a later-generation quinolone, an injectable, and the regimen was further added with three second-line anti-TB drugs [17]. Treatment success for MDR/RR-TB was around 60% and only 26% for XDR-TB [18]. However, the last decade showed considerable developments in the treatment of MDR/RR-TB. Three new anti-TB drugs became available in the last decade: bedaquiline, delamanid, and pretomanid. Furthermore, treatment duration could be shortened significantly with increased treatment success, making use of two of these three drugs.

The first studies with bedaquiline, a diarylquinoline, showed improved success rates when added to a treatment regimen for MDR/RR-TB [19,20,21]. This led to accelerated approval of the U.S. Food and Drug Administration (FDA) in 2012 and a conditional approval of the EMA in 2014. After some years, large real-life studies also showed increased favorable success percentages of 71.3% and 73% for bedaquiline-containing regimens [22,23].

Delamanid, a bicyclic nitroimidazole, was approved by the EMA in 2014 for adult pulmonary TB if another appropriate regimen could not be comprised. The phase-3 trial with delamanid showed a favorable success percentage of 74.5% [24]. Due to concerns on QTc prolongation of both bedaquiline and delamanid, the first studies, when combining both drugs at the same time or using them consecutively, mostly looked at safety. One early case series looking at the combination of bedaquiline and delamanid showed that out of five patients, one was cured, and three had culture conversions, while only two patients had QTc prolongation [25]. Another small study that also looked at efficacy showed that 74% of the patients had culture conversion after 6 months, while there was not an increased risk of QTc prolongation [26].

In 2018, a large meta-analysis using individual patient data, coordinated by the group from McGill University [5], showed that linezolid, moxifloxacin, levofloxacin, bedaquiline, clofazimine, and carbapenems were positively associated with treatment success. Linezolid, bedaquiline, moxifloxacin, and levofloxacin were also associated with reduced deaths. Of the injectables, only amikacin provided modest benefits. Based on this meta-analysis, the WHO provided an updated guideline in 2019 [27]. As studies with delamanid were scarce, this drug could not be classified well.

The last new anti-TB drug, pretomanid, was approved for treatment in MDR or (pre-)XDR-TB by the FDA in 2019 and by the EMA in 2020. The first studies were all looking at the (early) bactericidal activity of pretomanid after 2 or 8 weeks [28]. The most recent studies on treatment for MDR-TB and (pre-)XDR-TB consist of regimens with bedaquiline, pretomanid, and linezolid, the so-called BPaL regimen [6,7]. One study also looked at the addition of moxifloxacin or clofazimine to this BPaL regimen in comparison with the BPaL regimen [29]. The duration of the BPaL(M) regimen is only 6 months compared to the 9 months of the previous short-course MDR-TB regimen and the 18-month long-course MDR-TB regimen. Next to the improvement in the duration, the increase in effectiveness is also striking. Percentages of favorable outcomes were between 77% and 93% [7,29]. However, the treatment success percentages in the studies with the BPaL regimen depended on the dose and duration of linezolid, with the 1200 mg dose showing higher treatment success but also a higher incidence of adverse events than the 600 mg dose, and 26 weeks of therapy showed higher favorable outcomes but higher adverse events than 9 weeks [7]. The study looking at BPaLM and BPaLC showed success percentages of 89% and 81%, respectively [29].

Currently, the main challenge is that only an estimated 43% of MDR/RR-TB patients receive appropriate treatment worldwide [1], and this should be urgently addressed. This means that fast access to both DST and second-line drugs is of imminent importance. Both the high cost and availability of bedaquiline, pretomanid, and linezolid limit access, not only in low-resource countries but also in middle- and high-income countries [30]. Another main concern is the increasing resistance against bedaquiline. In a small study in South Africa, the baseline resistance against bedaquiline was 8%. The same study showed that the acquired resistance against bedaquiline was 47% [8]. Baseline resistance can be detected with better access to DST, and acquired resistance shows there is a need for therapeutic drug monitoring for bedaquiline or for a standardized higher dose [31].

## 3. Current Drug Susceptibility Testing Methods

The main tests for drug susceptibility detection in *Mycobacterium tuberculosis* complex (MTBC) can be currently classified into phenotypic and genotypic (molecular) tests. In phenotypic testing, culture on liquid media is still considered the gold standard for MTBC detection confirmation as it allows for the detection of as few as 10 viable bacilli per ml [32,33], and it is the essential first step to implement phenotypic drug susceptibility testing (pDST). Nevertheless, this diagnostic technique requires appropriate infrastructures, biosafety conditions, and suitable training to be performed. Moreover, the associated costs constitute a real challenge for its wider implementation in LMICs, where the TB burden is still at its highest. Finally, even where costs and operational requirements do not represent a limiting factor for the implementation of culture, it is important to state that 1 to 3 weeks are necessary to have a culture-positive sample. This long period of incubation, needed to ensure MTBC growth, is a main factor affecting the overall TB care turnaround time [33].

To overcome the limits of the phenotypic tests, the WHO currently recommends the use in all settings of molecular WHO-recommended diagnostics (mWRDs)/genotypic tests for both the initial TB diagnosis and for at least rifampicin resistance detection, with the final goal of minimizing delays in the administration of the most appropriate treatment [33]. Figure 1 illustrates the phenotypic and genotypic tests available for DST. Among the mWRDs, the fully automated cartridge-based tests Xpert MTB/RIF and Xpert MTB/RIF Ultra assays (Cepheid, Sunnyvale, CA, USA) allow for the identification of MTBC and rifampicin resistance directly from sputum in less than 2 h and have a limit of detection between 15 and 150 bacilli/mL [32]. Other assays with similar characteristics and performance in MTBC and rifampicin resistance detection are the Truenat MTB, MTB Plus, and MTB-RIF Dx assays (Molbio Diagnostics, Verna, Goa, India), chip-based real-time micro-PCR assays [34,35]. Because of the minimal training needed as well as limited infrastructure requirements, these assays have been implemented in several peripheral settings, favoring the decentralization of TB diagnostics with the main aim of increasing access to fast and reliable TB diagnostic tools.

In addition, detecting isoniazid-resistant rifampicin-susceptible TB (Hr-TB) is also essential, as Hr-TB is the most prevalent type of DR-TB, and when Hr-TB is undetected and patients are treated with first-line TB drugs, the likelihood of unfavorable treatment outcomes increases [36]. The mWRDs that can simultaneously detect MTBC and resistance to both rifampicin and isoniazid are line probe assays (LPA), moderate-complexity automated NAATs such as BD MAX MDR-TB (Franklin Lakes, NJ, USA), cobas MTB-RIF/INH (Roche Diagnostics, Rotkreuz, Switzerland), FluoroType MTBDR (Hain Lifescience, Nehren, Germany), and Abbott RealTime MTB RIF/INH (Abbott Park, IL, USA), or low-complexity NAATs such as Xpert MTB/XDR assays (Cepheid, Sunnyvale, CA, USA) [37] and multidrug-resistant loop-mediated isothermal amplification (MDR-LAMP, Eiken Chemical, Tokyo, Japan) [38]. However, the WHO recommendation to scale up new shorter regimens for DR-TB, along with the revised definition of XDR-TB, has necessitated the expansion of testing beyond rifampicin and isoniazid to include fluoroquinolones, bedaquiline, pretomanid, and linezolid [39,40].

In this scenario, culture-based phenotypic (p)DST is still considered the gold standard for determining resistance to these drugs [12,13]. Aside from the gold standard pDST in mycobacterial growth indicator tube or Middlebrook 7H11 media, another method for performing pDST is the determination of Minimum Inhibitory Concentrations (MIC) testing using the broth microdilution method in 96-wells plates [13]. This method offers several advantages, including the possibility to characterize novel resistance mutations detected by sequencing. Therefore, it can play a major role in clinical trials evaluating new regimens as well as in surveillance [41]. Nonetheless, pDST has proved to be technically challenging, costly and time-consuming.

Therefore, molecular tests (genotypic (g)DST) able to predict drug resistance also to the newly recommended shorter regimen drugs are deeply needed to guarantee DR-TB care and control. While new rapid molecular tools have recently been put on the market, these can predict resistance only to fluoroquinolones, second-line injectable drugs, and ethionamide [14]. To date, only NGS has the potential to predict resistance to all groups of anti-TB medicines. The WHO’s recently released and updated handbook on TB molecular diagnostics further supports the use of targeted (t)NGS for TB resistance detection [33]. It allows for the concurrent identification of resistance to multiple drugs and offers faster direct testing on clinical samples, bypassing the need for culture-based testing (3–5 days vs. 4–6 weeks, respectively) [42]. If capacity is built at the central level in high-burden countries, targeted NGS (tNGS) may support the rapid detection of group A drugs’ resistance, in particular, bedaquiline and linezolid.

tNGS commercially available kits, such as Deeplex MycTB by GenoScreen, can provide a comprehensive profile of TB drug resistance, including bedaquiline, clofazimine, and linezolid [43]. This kit is run on the Illumina platform. The other available approach for tNGS is based on Oxford Nanopore technology, currently WHO-endorsed only for bedaquiline, although the range of available technologies and their targets is rapidly evolving [33]. Paired with recently developed tools such as the WHO Catalogue of mutations in Mycobacterium tuberculosis complex and their association with drug resistance, whose new version has been released, can allow for a rapid and straightforward interpretation of the main known mutations associated with resistance [44].

## 4. The Gaps and Challenges in the DST Implementation

Among individuals with bacteriologically confirmed pulmonary TB, only 73% underwent testing for rifampicin resistance [1]. The availability of pDST remains limited even in Europe [45]. pDST poses technical challenges and is costly and time-consuming. Among the mWRDs, NAATs and LPA have limitations in predicting resistance to two group A drugs, i.e., bedaquiline and linezolid [14]. Other mWRDs, i.e., whole-genome sequencing (WGS) and NGS or tNGS, face challenges in delivering effective results in many LMICs due to limited infrastructure, such as data warehouses or cloud storage [15]. Our understanding of the molecular mechanisms underlying drug resistance emergence is also incomplete, restricting the predictive capabilities of NGS [46].

Sequencing facilities in LMICs are often more expensive than in high-income countries. This is attributed to increased costs related to shipments, customs, and higher prices charged by regional or national distributors [16]. Furthermore, countries without national distributors encounter challenges in procurement and supply chains, hindering optimal maintenance, support, importation, and transportation of samples or reagents [46]. The availability of supplies is further hampered by bureaucratic and time-consuming procedures associated with the importation of donated goods [46]. Substantial degradation in materials’ shelf life occurs during transportation from the manufacturer to the distributor and, subsequently, to the implementation site [46]. Moreover, many LMICs lack the expertise of skilled human resources to process and analyze sequencing outputs [47]. Additional challenges include problems with installation, the complexity of protocols and workflows, not yet fully standardized data management, and unreliable internet connectivity [42,46]. Table 1 shows the current DST methods and their application in LMICs.

## 5. Strategies for Achieving Equitable DST Access and Gap Closure

Ensuring equitable access and narrowing the gap in DST worldwide is an important step towards achieving TB elimination. Below, we explore strategies aimed at enhancing diagnostic capabilities, especially for emerging DR-TB drugs, and fostering equitable access to DST.

### 5.1. Strengthening the Health System

Strengthening the health system is foundational for successful improved access to DST. Public health interventions should be focused on strengthening laboratory infrastructure, healthcare workforce, and quality assurance programs. Adequate laboratory infrastructure is necessary for conducting accurate and timely diagnostic tests for TB. A well-trained and sufficient healthcare workforce is essential for the proper implementation of TB diagnostic and treatment protocols. This includes not only laboratory technicians proficient in conducting DST but also healthcare providers who can interpret the results and initiate appropriate treatment. Quality assurance programs help ensure the accuracy and reliability of laboratory tests. This involves implementing quality control measures, proficiency testing, and ongoing training and supervision to maintain high standards of laboratory performance. By investing in these aspects, health systems can efficiently diagnose and treat MDR/RR-TB cases [61]. This investment may involve utilizing domestic and international funding [1] and requires political commitment to support the availability of tools and materials [1,46]. Notably, international donor funding remains vital for LMICs, constituting 52% of available funding in 2022 within the 26 high-TB-burden countries and two global TB watchlist countries (Cambodia and Zimbabwe) outside Brazil, Russia, India, China, and South Africa. In low-income countries (LICs) specifically, it accounted for 61% of available funding [1].

### 5.2. Sharing Infrastructure and Human Resources

To streamline sequencing costs, countries may opt to establish a national or regional referral hub for sequencing TB strains, with shared infrastructure and human resources [15], or collaborate and pool resources with other infection control efforts, such as malaria, COVID-19, or antimicrobial resistance, for surveillance, control, and research purposes [62,63,64]. By centralizing sequencing efforts, duplication of equipment and efforts can be minimized, leading to cost savings. Additionally, pooling resources in a referral hub can facilitate collaboration and knowledge sharing among researchers, ultimately enhancing the overall effectiveness of sequencing initiatives at a national or regional level. Meanwhile, collaborating and pooling resources with other infection control efforts enables a more comprehensive and coordinated response to infectious diseases.

### 5.3. Providing DST Machine and Reagents at Subsidized or Reduced Prices

Global Fund support plays a pivotal role in providing necessary financial resources for drug procurement and distribution, ensuring availability in LMICs. Subsidized pricing strategies offered by pharmaceutical companies and facilitated through licensing agreements further enhance accessibility, as recently proven for bedaquiline [65]. In the realm of sequencing methods, advocating for a reduction in machine and reagent costs and opting for tNGS instead of WGS may help reduce expenses [15,66]. An alternative approach involves modifying pricing structures or advocating for loans with favorable, low interest rates [67].

### 5.4. Involvement of Drug Developers in the Development of DST Methods

Involving drug developers in the development of DST methods is noteworthy [68], as the prolonged efficacy of their drugs aligns with their long-term interests. In addition, by actively participating in the development of testing protocols, drug developers can also contribute their insights into the characteristics and behaviors of their compounds.

### 5.5. Capacity-Building Initiatives

Concurrently, capacity-building initiatives will empower laboratory staff and healthcare professionals, equipping them with the knowledge and skills required for the effective implementation and monitoring of DST technologies [15,69]. This can be achieved through ongoing mentorship and support, train-the-trainer initiatives, internships, knowledge transfer programs, and the utilization of e-learning platforms [70].

### 5.6. Maximizing the Utilization Efficiency of Sequencing Facilities

The sequencing facilities need to be utilized to their maximum potential to maximize the return on investment. This can be achieved through careful planning of both wet and dry infrastructures associated with NGS [46]. Wet infrastructure involves laboratory and experimental aspects, while dry infrastructure refers to computational and bioinformatics resources. Additionally, implementing a human resources investment plan [46], with thoughtful allocation of skilled personnel, including scientists, technicians, and bioinformaticians, can ensure the smooth operation of sequencing facilities.

To facilitate the implementation of sequencing, countries may consider forming a Clinical Advisory Committee composed of representatives from various entities, including the National Tuberculosis Program, National TB Reference Laboratories, laboratory specialists, and clinicians [46]. This committee would oversee the review and discussion of reports generated by tNGS, providing guidance to clinicians based on the findings. Additionally, adopting the end-to-end user-friendly NGS solution or remote technical support can help solve problems that arise during the NGS process [46].

Afterward, regular monitoring and evaluation of sequence implementation need to be carried out to evaluate the performance of the sequencing facilities [46]. This involves assessing the efficiency, accuracy, and throughput of the sequencing processes. Through systematic evaluation, any shortcomings or areas needing enhancement can be identified, which allows for targeted improvements, ensuring that the facilities meet the technological advancements and scientific requirements. To enhance the utilization of sequencing facilities, a country can implement compulsory molecular surveillance monitoring for drug resistance [68]. This compulsory surveillance will also enable the country to stay ahead of emerging resistance patterns and guide appropriate adjustments in treatment protocols.

### 5.7. Encouraging Public–Private Partnerships

Public–private partnerships are essential to leveraging resources and expertise to expand access to DST services in underserved areas, especially for high-burden TB countries that have large private health sectors [71]. The public and private sectors bring different strengths to the table. The public sector often has experience in managing large-scale healthcare programs and expertise in public health policy, while the private sector may offer innovation, efficiency, and specialized medical services. Collaborating through public–private partnerships allows for the sharing of knowledge and best practices, leading to more effective TB control strategies. Public–private partnerships often involve community-based organizations and civil society groups in their initiatives. This engagement is essential for raising awareness about TB, promoting early detection, reducing stigma, and identifying as well as addressing barriers to accessing TB services in underserved areas. Engaging private laboratories in Nigeria has led to an increase in access to Gene Xpert testing and subsequent notification of TB cases [71].

### 5.8. Recognizing Successful Case Studies or Initiatives

Recognition of successful case studies or initiatives that have demonstrated progress can motivate and guide future efforts to narrow the gaps in DST access, such as the development of an implementation strategy in Namibia and a diagnostics delivery model using an Academic Model Providing Access to Healthcare (AMPATH) vehicles and re-tracking policy in Kenya [46,69]. The AMPATH vehicles make regular visits to health facilities, typically ranging from one to four times per week, transporting supplies, specimens, and personnel. Meanwhile, the re-tracking policy is in the form of an algorithm that is used to identify results that are delayed or possibly lost. These efforts have resulted in a significant increase in the proportion of cases receiving DST in Kenya, up to 24.7% [69].

### 5.9. Promoting Research and Development

To address the challenges associated with DST methods, it is imperative to foster research and development [1], including establishing clear-cut values of pDST for both new and repurposed drugs and engaging drug developers in the process of developing DST methods [68]. Countries should also implement stringent quality assurance measures to maintain the accuracy and reliability of DST results, including regular proficiency testing and external quality assessments [45]. Meanwhile, to facilitate NGS implementation in countries with hot climates, there is a need to develop reagents/technologies that do not require temperature control [46].

### 5.10. Initiatives for Technology Transfer

Technology transfer initiatives can empower researchers in LICs engaged in basic science research [72], enabling them to develop their own sequencing devices or materials, thereby fostering self-sustainable NGS capacity. This reduces dependence on external sources for technology and supplies, making DST more accessible and affordable in the long run. Moreover, technology transfer initiatives can stimulate innovation and entrepreneurship in LICs by encouraging researchers to adapt and improve existing sequencing technologies or develop new solutions tailored to the specific needs and challenges of their region. Consequently, LICs can address their local health challenges more effectively, ultimately contributing to improvements in public health outcomes.

We summarize the challenges in the DST methods and implementations and outline the strategies to overcome them in Table 2.

To ensure the efficacy of measures against DR-TB, it is necessary to continually assess the extent of their impact on public health worldwide. This entails analyzing various factors, such as their effects on access to TB testing, notification of TB cases, and the overall health systems.

## 6. Conclusions

Detecting resistance to new anti-TB drugs is crucial to maximize the effectiveness of new therapeutic regimens and prevent the depletion of these valuable weapons in the fight against DR-TB. Although several challenges exist in DST methods and their implementation, there are opportunities for innovation and improvement. Through the implementation of various appropriate strategies, we could enhance diagnostic capabilities for new DR-TB drugs and promote equal access to DST, thereby contributing to global efforts to combat this formidable public health threat. Evaluating the extent of the impact of these measures is essential to ensure their efficacy against DR-TB.

## Figures and Tables

**Figure 1 pathogens-13-01045-f001:**
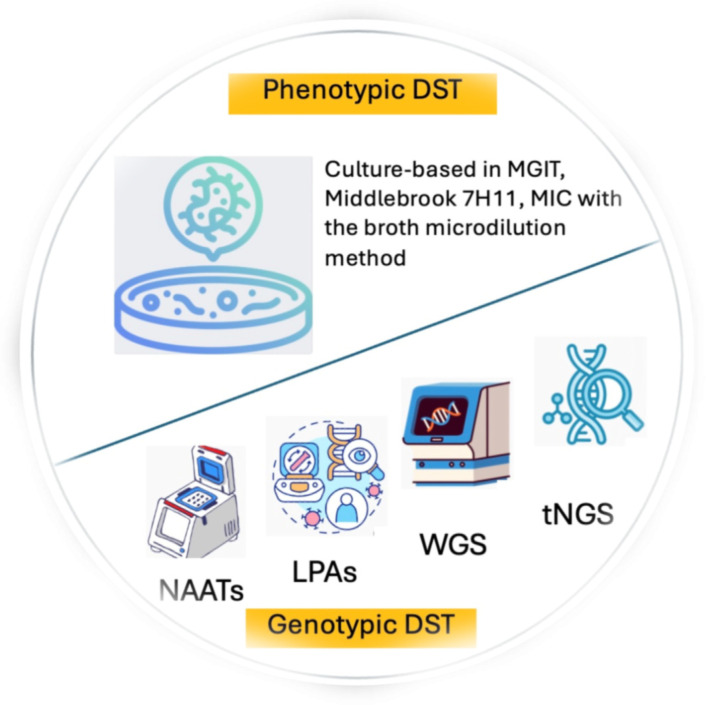
Phenotypic and genotypic tests available for DST.

**Table 1 pathogens-13-01045-t001:** Current DST methods and their use in LMICs.

	Turnaround Time [48,49,50,51,52]	Sensitivity and Specificity [32,53,54,55]	Complexity of Procedure [15,33]	Complexity of Infrastructure [15,42]	Cost [16,56,57,58]	The Use in LMICs, Including the Setting [33,59,60]	Examples of Platforms [15,38]
Phenotypic DST							
Culture-based	Long (1–3 weeks for liquid culture, 4–6 weeks for solid culture)	Highly sensitive and specific (limit of detection: 10 CFU/mL)	Moderate, need suitable training	Moderate, need biosafety conditions	Moderate to high (21.5–119 USD)	Used as a reference test, usually in tertiary health centers/referral laboratories	Culture-based in MGIT, Middlebrook 7H11, MIC with the broth microdilution method
Genotypic DST							
NAATs	Short (1–3 h, can extend up to 2 days if there are delays in sample shipment or result delivery)	Highly sensitive and specific (limit of detection: 15–150 CFU/mL)	Low to moderate	Low	Low (13.8 USD)	Used as an initial or confirmation test, usually in secondary health centers/laboratories	Xpert MTB/RIF and Xpert MTB/RIF Ultra (Cepheid, Sunnyvale, CA, USA); Truenat (Molbio, Verna, Goa, India); Abbott RealTime MTB and Abbott RealTime MTB RIF/INH (Abbott, Des Plaines, IL, USA); BD MAX MDR-TB (Becton Dickinson, Franklin Lakes, NJ, USA); cobas MTB and cobas MTB-RIF/INH (Roche, Basel, Switzerland); FluoroType MTBDR and FluoroType MTB (Hain Lifescience/Bruker, Tübingen, Germany); MDR-LAMP (Eiken, Tokyo, Japan)
LPAs	Short (5 h, can extend up to 2 days if there are delays in sample shipment or result delivery)	Highly sensitive and specific (limit of detection: 10,000 CFU/mL)	Moderate to high, need multiple steps	Moderate to high, need separate rooms for different steps	Low (18.6 USD)	Used as an initial or confirmation test, usually in secondary health centers/laboratories	GenoType MTBDRplus v1 and v2, and GenoType MTBDRsl (Hain Lifescience/Bruker, Tübingen, Germany); Genoscholar NTM + MDRTB II, and Genoscholar PZA-TB II (Nipro, Mechelen, Belgium)
WGS	Long (6–11 days, can extend up to 25 days if there are delays in sample shipment or result delivery)	Highly sensitive and specific	High, need culture prior to WGS and expertise of skilled human resources to process and analyze sequencing outputs	High, need appropriate installation, procurement, and supply chains, as well as reliable internet connectivity	High (141–277 USD)	Used as a confirmation test, in tertiary health centers/referral laboratories. WGS is also useful for surveillance and source investigation	Miseq, MiniSeq, NextSeq, HiSeq (Illumina, San Diego, CA, USA); Ion Personal Genome Machine Sequencer (Thermo Fisher Scientific, Waltham, MA, USA); PacBio RS II (Pacific Biosciences, Menlo Park, CA, USA); MinION (Oxford Nanopore Technologies, Oxford, UK)
(t)NGS	Short (2–3 days, can extend up to 10 days if there are delays in sample shipment or result delivery))	Highly sensitive and specific (limit of detection: 100 CFU/mL)	High, need skilled human personnel, but can be used directly on clinical specimens	High, similar to WGS	High, but less than WGS (78.3–230 USD)	Used as a confirmation test, in tertiary health centers/referral laboratories	Same as WGS

DST: drug sensitivity testing, NAATs: nucleic acid amplification tests, LPAs: line probe assays, WGS: whole-genome sequencing, (t)NGS: (targeted) next-generation sequencing, CFU: colony forming unit, LMICs: low and middle-income countries, and MIC: minimum inhibitory concentration.

**Table 2 pathogens-13-01045-t002:** Challenge and proposed solutions for the DST methods and implementations.

Challenges	Proposed Solutions
Health system	
Limited tools and infrastructure, supply chain challenges, sustainability	Investing in infrastructure, with domestic and international funding [1] Establish a regional referral hub for sequencing TB strains, with shared infrastructure and human resources [15] Collaborate and pool resources with other infection control efforts [62,63,64] Development of temperature-insensitive technologies/materials [46] Political commitment to support the availability of tools and materials [1,46] Technology transfer initiatives [72]
High cost (capital investment, running, data storage, and overhead expenses)	Advocating for lowering costs [15], altered pricing structures or the provisions of loans with low interest rates [67] Involving the drug developers in the DST development [68] Investing in infrastructure, with domestic and international funding [1]
Lack of expertise (bioinformatics, clinical interpretation)	Capacity building for laboratory staff and healthcare professionals [15,70] Establishment of Clinical Advisory Committee [46] End-to-end user-friendly NGS solutions [46] Remote technical support [46]
Inefficient use of the sequencing capacity, limited coverage	Develop careful plans for NGS infrastructure construction and human resources investment, followed by routine monitoring and evaluation [46] Mandatory molecular surveillance monitoring for drug resistance [68] Establish a national referral hub where all samples undergo processing [15] Highlighting and acknowledging successful case studies or initiatives that have made progress [46,69] Expanding testing access in underserved areas through public–private partnerships [71]
DST methods	
Lack of clear-cut values for new TB drugs (pDST), some resistance mechanisms cannot be explored, difficulty interpreting whole-genome variation data when a significant number of rare variants are present (gDST)	Fostering research and development in DST methods, particularly for key drugs in DR-TB regimens [1] Involving the drug developers in the DST development [68]
Inappropriate use of DST	Implement stringent quality assurance measures [45]

## Data Availability

The original contributions presented in this study are included in the article. Further inquiries can be directed to the corresponding author.

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
