# Peer review of "Strategies to Enhance Diagnostic Capabilities for the New Drug-Resistant Tuberculosis (DR-TB) Drugs"

_pathogens, 2024, doi:10.3390/pathogens13121045_

Round 1
Reviewer 1 Report
Comments and Suggestions for Authors
The authors described the measures used to combat DR-TB, focusing on identifying resistance of new drugs against DR-Tb.
The paper is well written, but the argument is more difficult to understand, so I suggest adding in the part where the authors describe the current diagnostic methods a figure summarizing the chapter.
Also, all acronyms such as QTc and Mycobacterium tuberculosis should be written in italics.
Author Response
|
Point-by-point response to comments and suggestions from the reviewers
Reviewer 1: The authors described the measures used to combat DR-TB, focusing on identifying resistance of new drugs against DR-Tb. The paper is well written, but the argument is more difficult to understand, so I suggest adding in the part where the authors describe the current diagnostic methods a figure summarizing the chapter. Also, all acronyms such as QTc and Mycobacterium tuberculosis should be written in italics.
|
|
Response 1: Thank you for pointing this out. We agree with your suggestion and have addressed it by creating a figure that summarizes the current drug susceptibility methods discussed in the chapter, as shown below:
Figure 1. Phenotypic and genetic tests available for DST (please see the attachment) Furthermore, to align the chapter with the manuscript’s focus on drug susceptibility methods, we have removed the section on sputum smear examination, as it cannot be used to detect resistance to TB drugs. This revision, including the new figure, can be found on page 4, line 149 -176 of the revised manuscript. In addition, we have revised all acronyms and formatted QTc and Mycobacterium tuberculosis in italics throughout the text. These changes can be found on page 3, line 106, 110, and 112, page 4, line 149, and page 5, line 254 of the revised manuscript. We also updated our references to reflect the recently release of the WHO’s updated handbook module 3 on TB molecular diagnostic (references 43), and updated that cut-off values for pretomanid in pDST are now available, resulting in the removal of corresponding sentences (page 1 line 69, page 5 line 234, and page 5 line 260) and removal of reference 15.
|
Reviewer 2 Report
Comments and Suggestions for Authors
I really enjoyed reading and reviewing this well-written review. The ideas presented in Section 5 regarding strategies for achieving equitable DST access and gap closure are thought-provoking. I congratulate the authors on articulating such strong concepts to improve drug resistance screening for M. tuberculosis infections.
Aside from some minor changes, I find the article well-written and easy to read. Please find the attached summary of comments, which includes suggestions for adding a few citations to support your statements.

Author Response
Thank you for your positive feedback and valuable suggestions. We have, accordingly, added several citations to support the statements, which can be found on page 2 line 64 of the revised manuscript. For your convenience, we have included the references below:
- Derendinger B, Dippenaar A, de Vos M, Alberts R, Sirgel F, Dolby T, et al. High frequency of bedaquiline resistance in programmatically treated drug-resistant TB patients with sustained culture-positivity in Cape Town, South Africa. Int J Mycobacteriology. 2021;
- Millard J, Rimmer S, Nimmo C, O’Donnell M. Therapeutic Failure and Acquired Bedaquiline and Delamanid Resistance in Treatment of Drug-Resistant TB. Emerg Infect Dis. 2023;
- He W, Liu C, Liu D, Ma A, Song Y, He P, et al. Prevalence of Mycobacterium tuberculosis resistant to bedaquiline and delamanid in China. J Glob Antimicrob Resist. 2021;
- Arora G, Bothra A, Prosser G, Arora K, Sajid A. Role of post-translational modifications in the acquisition of drug resistance in Mycobacterium tuberculosis. FEBS Journal. 2021.
Additionally, we have re-written Mycobacterium tuberculosis in italics, that can be found on page 4, line 149, and page 5, line 254.
Reviewer 3 Report
Comments and Suggestions for Authors
The manuscript "Strategies to Enhance Diagnostic Capabilities for the New Drug-Resistant Tuberculosis (DR-TB) Drugs" is well written. However, few points needs to be added to the present version
1. To add a note on LAMP and LAM test which are also WHO endorsed test
2. There is a need to distinguish initial and follow on diagnostic test.
3. Please add a note on turn around time for these tests
4. Clearly dfferentiate NGS from pDST. Please add a note on difference between WGS and tNGS and how it is useful in LMC and in which setting
5. The table on challenges and proposed solution fails to cover NGS. if It is strictly phenotypic DST please change the title
Comments on the Quality of English Language
None
Author Response
|
Reviewer 3: The manuscript "Strategies to Enhance Diagnostic Capabilities for the New Drug-Resistant Tuberculosis (DR-TB) Drugs" is well written. However, few points needs to be added to the present version 1. To add a note on LAMP and LAM test which are also WHO endorsed test 2. There is a need to distinguish initial and follow on diagnostic test. 3. Please add a note on turn around time for these tests 4. Clearly dfferentiate NGS from pDST. Please add a note on difference between WGS and tNGS and how it is useful in LMC and in which setting 5. The table on challenges and proposed solution fails to cover NGS. if It is strictly phenotypic DST please change the title
Response 3: We sincerely thank the reviewer for his/her meticulous feedback. Based on these inputs, we have made several changes in the revised manuscript, detailed as follows:
1. We have added the LAMP test to Chapter 3, “Current Drug Susceptibility Testing Methods” (page 5, line 222-223) and included it in Table 1, “Current DST methods and their use in LMICs” (page 6-8, line 328-331). Additionally, to maintain the focus on diagnostic tests that can also serve as drug susceptibility testing, we have removed tests that cannot detect resistance to TB drugs, such as sputum smear examination. Therefore, we did not add LAM test to the manuscript, as it does not detect resistance to TB drugs.
2. We have differentiated Drug Susceptibitily Tests (DST) into initial and follow up/confirmation tests, as outlined in Table 1, “Current DST methods and their use in LMICs” (page 6-8, line 328-331). We have attached the table here, for your reference.
Table 1. Current DST methods and their use in LMICs
3. Thank you for your valuable input. We have included the turn-around-time for the DST methods in Table 1, “Current DST methods and their use in LMICs” (page 6-8, line 328-331), which can also be found above.
4. We have differentiated NGS from the phenotypic(p)DST in the table as well. We wrote that NGS is part of genetic DST, offering a shorter turn-around-time compared to pDST, but with greater procedural complexity, infrastructure requirements, and cost. Additionally, we wrote down the differences in platforms for pDST and NGS. Furthermore, we highlighted that WGS has a higher cost and longer turn-around time than tNGS since WGS requires a culture to obtain adequate speciments. Their implementation in LMICs is also detailed in the table, indicating that WGS and tNGS are typically used as confirmation tests in tertiary health centers or referral laboratories due to their procedural complexity and infrastructure requirement. WGS is also useful in LMIC for surveillance and source investigations.
5. In the Table 2 (page 12), we have added information on the challenges associated with NGS to provide a more comprehensive review. In addition to the inability to explore certain resistance mechanisms, we noted the difficulty in interpreting whole-genome variation data when a significant number of rare variants are present. The proposed solutions to address these challenges remain the same: fostering research and development in DST methods, particularly for key drugs in DR-TB regimens, and involving the drug developers in the DST development. We have included these changes in the table here, for your reference.
|
||||||||||||||||||||||||||||||||||||||||||||||||||||||||||||||||||||||
Round 2
Reviewer 3 Report
Comments and Suggestions for Authors
Thank you for the revisions